# Transcriptome Analysis of Cabbage Near-Isogenic Lines Reveals the Involvement of the Plant Defensin Gene *PDF1.2* in *Fusarium* Wilt Resistance

**DOI:** 10.3390/ijms26083770

**Published:** 2025-04-16

**Authors:** Cunbao Zhao, Xing Liu, Ailing Zhou, Jialei Ji, Yong Wang, Mu Zhuang, Yangyong Zhang, Limei Yang, Lisong Ma, Biju V. Chellappan, Anna M. Artemyeva, Honghao Lv

**Affiliations:** 1State Key Laboratory of Vegetable Biobreeding, Institute of Vegetables and Flowers, Chinese Academy of Agricultural Sciences, Beijing 100081, China; cunbaozhao@163.com (C.Z.); liuxing02@caas.cn (X.L.); ailingzhou0518@163.com (A.Z.); jijialei@caas.cn (J.J.); wangyong03@caas.cn (Y.W.); zhuangmu@caas.cn (M.Z.); zhangyangyong@caas.cn (Y.Z.); yanglimei@caas.cn (L.Y.); 2State Key Laboratory of North China Crop Improvement and Regulation, College of Horticulture, Hebei Agricultural University, Baoding 071001, China; 3Department of Biological Science, College of Science, King Faisal University, Hofuf 31982, Saudi Arabia; bchellappan@kfu.edu.sa; 4Federal Research Center N.I. Vavilov All-Russian Institute of Plant Genetic Resources, 190000 St. Petersburg, Russia; akmell@yandex.ru

**Keywords:** cabbage, near-isogenic lines, *Fusarium oxysporum* f. sp. *conglutinans*, plant hormones, plant defensin protein

## Abstract

*Fusarium* wilt of cabbage (*Brassica oleracea* var. *capitata*), caused by *Fusarium oxysporum* f. sp. *conglutinans* (*Foc*), poses a significant threat to global cabbage production. Although resistance screening and the initial cloning of resistance genes in cabbage have been previously reported, the specific molecular mechanisms underlying cabbage resistance to *Foc* remain largely unknown. To elucidate the underlying mechanisms, we performed RNA sequencing analysis on a near-isogenic resistant line YR01_20 and a susceptible NIL line S01_20 by comparing both *Foc*-inoculated and mock-inoculated conditions. A total of 508.6 million sequencing raw reads (76.8 Gb data volume) were generated across all samples. Bioinformatics analysis of differentially expressed genes (DEGs) between S01_20 and YR01_20 revealed significant enrichment in plant hormone signaling and mitogen-activated protein kinase (MAPK) pathways. Notably, *BolC06g030650.2J*, encoding the plant defensin protein PDF1.2, was significantly upregulated in both pathways. Real-time quantitative PCR (RT-qPCR) analysis confirmed that *PDF1.2* was significantly upregulated in the resistant line at 12 h post-inoculation and remained elevated for up to 144 h. Furthermore, transgenic cabbage overexpressing *PDF1.2* exhibited significantly enhanced resistance to *Foc*. Taken together, these findings advance our understanding of the molecular mechanisms governing cabbage resistance to *Fusarium* wilt and identify *PDF1.2* as a genetic target for breeding *Foc*-resistant cabbage cultivars through molecular approaches.

## 1. Introduction

Cabbage (*Brassica oleracea* var. *capitata*), a major vegetable crop within the Brassicaceae family, is widely cultivated worldwide. Cabbage Fusarium Wilt (CFW), caused by *Fusarium oxysporum* f. sp. *conglutinans* (*Foc*), is a soil-borne fungal disease that significantly impacts cabbage yield and quality [1]. Initially reported in the United States, CFW has subsequently spread across major cabbage-growing regions in the US, Europe, and Asia, emerging as a primary constraint to cabbage production [2]. In 2001, CFW was first reported in Yanqing, Beijing, China, and has since rapidly expanded to northern cabbage-growing regions, with recent occurrences in some southern areas [3]. *Foc* is classified into two physiological races, with race 1 being globally dominant, and all *Foc* isolates identified in China thus far belong to this race [1,4]. Given the severe threat posed by CFW, extensive research has focused on resistance genetics, gene cloning, and molecular marker development to support breeding for resistance. Previous studies have demonstrated that cabbage varieties, such as 96–100, exhibit stable resistance to race 1, with this resistance being consistent across diverse environmental conditions and following a dominant monogenic inheritance pattern [5]. Through map-based cloning, the *FOC1* resistance gene in cabbage was characterized, and a series of efficient molecular markers were developed to facilitate breeding for *Foc* resistance [6]. Subsequently, using diverse cabbage and Chinese cabbage (*Brassica rapa*) germplasms, *FocBo1*/*FocBr1* was confirmed to be isogenic with *FOC1* [7,8].

Plant immune system is composed of two lines of defense: upon pathogen invasion, pattern recognition receptors (PRRs) on the plant cell surface recognize pathogen-associated molecular patterns (PAMPs), initiating the first line of defense, known as PAMP-triggered immunity (PTI). However, pathogens can secrete effector proteins to suppress PTI, leading to the activation of the second line of defense, known as effector-triggered immunity (ETI) [9]. In ETI, plant resistance (R) proteins recognize pathogen effectors, often triggering localized programmed cell death, known as the hypersensitive response (HR). The evolutionary arms race between plants and pathogens drives the continuous development of novel resistance mechanisms [9,10,11,12]. Resistance (*R*) genes play a pivotal role in regulating plant resistance. The majority of cloned *R* genes encode nucleotide-binding site-leucine-rich repeat (NBS-LRR or NLR) proteins, receptor-like kinases (RLKs), and others, with NLRs-classified into coiled-coil NLRs (CNLs) and Toll/interleukin-1 receptor NLRs (TNLs)-accounting for over 80% of characterized resistance proteins [13].

The *Fusarium* wilt resistance genes *Fom-1* and *FOC1* have been successfully cloned in melon (*Cucumis melo*) and cabbage, respectively, and both have been identified as TNL genes [6,14]. The activation of *R* genes triggers downstream defense signaling, involving the mitogen-activated protein kinase (MAPK) signaling pathway, transcription factors such as NAC, MYB, and WRKY, pathogenesis-related (*PR*) genes, and signaling pathways of hormones such as salicylic acid (SA), jasmonic acid (JA), ethylene (ET), gibberellins (GA), and abscisic acid (ABA), which function either independently or synergistically in plant defense responses [15,16,17,18].

Advancements in high-throughput sequencing technologies have provided powerful tools for investigating cabbage–*Foc* interactions, and for deciphering the molecular mechanisms underlying cabbage resistance to *Fusarium* wilt. Transcriptomic analysis of roots from the resistant cabbage variety R4P1 revealed distinct molecular signatures between *Foc*-inoculated samples and distilled water-treated controls, which are associated with significant activation of early defense systems, including the MAPK signaling pathway, calcium signaling, and SA-mediated hypersensitive responses following *Foc* inoculation [19]. Mining the cabbage genome has identified 105 TNL-type and 33 CNL-type resistance genes, with nine genes showing significant upregulation and five exhibiting marked downregulation upon *Foc* infection. Notably, *FOC1* appears to function in coordination with four additional clustered genes to confer resistance against *Foc* infection [20]. Furthermore, histopathological examinations demonstrated that root tissues of resistant germplasms effectively suppressed *Foc* hyphal penetration and colonization. Building on these findings, dual transcriptomic profiling of cabbage–*Foc* interaction revealed significant enrichment of plant–pathogen interaction pathways in the host and ribosome biogenesis pathways in the pathogen [1].

In this study, we conducted RNA sequencing and differential gene expression analysis on resistant near-isogenic cabbage line and susceptible cabbage line following *Foc* inoculation. Notably, the plant defensin gene *PDF1.2* exhibited pronounced upregulation specifically in resistant lines. The expression profile and functional analysis of *PDF1.2* demonstrated that *PDF1.2* is involved in the resistance against *Foc* in cabbage.

## 2. Results

### 2.1. RNA Sequencing of a Near-Isogenic Resistant Line Following Foc Inoculation

To dissect the mechanisms of *Fusarium* wilt resistance in cabbage, we conducted RNA-sequencing on a near-isogenic resistant line YR01_20 and a susceptible NIL line S01_20, following *Foc* inoculation at a series of hours post-inoculation (hpi), respectively. Comprehensive quality control metrics were assessed for both raw and processed sequencing data (Table 1). A total of 508.6 million raw reads were generated across all samples, yielding a sequencing output of 76.8 Gb. All samples achieved Q30 scores (base call accuracy ≥ 99.9%) of 93.94%, indicating high-quality sequencing data suitable for subsequent analyses. Following quality filtering to remove adapter sequences and low-quality reads, we obtained 471.5 million clean reads (71.2 Gb), accounting for 92.71% of the raw data, with a final data volume of 71.2 Gb. Base quality distribution plots (Appendix A) displayed consistently high-quality scores (>20) across all read positions, further confirming the integrity of the sequencing data.

Clean reads were mapped to the *B*. *oleracea* reference genome [21] using HISAT2 [22], achieving a mapping rate of 94.68% (Appendix A). 94.32% of the mapped reads exhibited unambiguous alignment (Appendix A). Chromosomal read distribution analysis indicated uniform coverage across all chromosomes (Appendix A). Gene structure mapping revealed that 87.98% of the reads aligned to genic regions, with 96.43% mapping to exonic regions (Appendix A), confirming high mapping specificity and ensuring data reliability for downstream analyses.

Transcript quantification was performed using HTSeq [23] with default parameters, generating raw read counts for each gene. Expression normalization was conducted using FPKM (Fragments Per Kilobase of transcript per Million mapped reads) [24] to account for transcript length and sequencing depth variations. Expression density analysis revealed a tripartite distribution pattern across all samples, with dominant expression clusters corresponding to non-expressed (0–0.01 FPKM), low expression (1–10 FPKM), and moderate expression (10–100 FPKM) ranges (Appendix A).

Transcript detection analysis showed consistent gene counts across biological replicates: S01_20_MOCK (31,001 ± 122), S01_20 inoculated (31,027 ± 57), YR01_20_MOCK (31,063 ± 80), and YR01_20 inoculated (31,094 ± 88) (mean ± SD; Figure 1A). Gene-set overlap analysis using UpSet visualization (Figure 1B) identified a core set of conserved transcripts and condition-specific gene clusters. Technical validation using pairwise Pearson correlation analysis revealed high inter-replicate consistency (r > 0.90) across most sample groups (Figure 1C), confirming the reproducibility of experimental results and the reliability of biological sample selection.

### 2.2. Analysis of DEGs in Cabbage

The analysis revealed infection-stage-specific differential patterns. The baseline comparison (mock-treated) identified 1,221 DEGs, comprising 465 upregulated and 756 downregulated genes. Upon *Foc* infection, the susceptible line exhibited 1,263 DEGs (856 upregulated, 407 downregulated), whereas the resistant line displayed a more robust transcriptional response with 1,415 DEGs (1,145 upregulated, 270 downregulated) (Figure 2A). Genotype comparison at 12 hpi identified 649 DEGs (364 upregulated and 285 downregulated) (Figure 2A). Volcano plot analysis showed the distribution of gene expression changes (Figure 2B), while a Venn diagram analysis identified 14 core responsive genes that were differentially expressed across all comparisons (Figure 2C). Additionally, hierarchical clustering analysis of all DEGs revealed distinct expression patterns across experimental conditions, with clear separation of transcriptional profiles (Figure 2D). Based on genomic information and RNA differential expression analysis results, genomic mapping of DEGs indicated that they were preferentially distributed on specific chromosomal regions exhibiting a higher density of infection-responsive genes (Appendix A).

### 2.3. Functional Enrichment Analysis of DEGs

Gene Ontology (GO) and Kyoto Encyclopedia of Genes and Genomes (KEGG) pathways analyses were employed to analyze the DEGs. GO enrichment analysis identified a three-tier defense mechanism: (i) cellular component analysis showed the downregulation of photosystem disassembly (GO:0009521, 8 DEGs ↓), (ii) molecular function highlighted the inhibition of geranylgeranyl reductase (GO:0045550, 2 DEGs ↓), and (iii) biological processes analysis demonstrated the suppression of photosynthesis (GO:0015979, 10 DEGs ↓/2 DEGs ↑) (Figure 3A and Appendix A). KEGG pathway analysis confirmed the downregulation of photosynthetic pathways (ko00195/ko00196/ko00860) and identified hormone signaling (ko04075, 11 DEGs) and MAPK cascades (ko04016, 8 DEGs) as top-enriched defense-related pathways (Figure 3B and Appendix A).

Given the well-established involvement of phytohormone and MAPK signaling in plant disease defense, comparative gene annotation was performed using BrassicaDB (http://brassicadb.cn/) and TAIR (https://www.arabidopsis.org/) reference databases to identify key regulatory elements (Table 2). The phytohormone signaling module exhibited a coordinated regulation pattern: (i) upregulated genes included two cytokinin signaling activators (*BolC03g044440.2J*, *BolC07g046020.2J*) and one ABA signaling repressor (PP2C, *BolC08g057620.2J*); (ii) downregulated components comprised three ABA receptors (PYR/PYL family: *BolC01g011300.2J*, *BolC03g024500.2J*, *BolC04g064360.2J*) and five hormone metabolism genes spanning auxin (*BolC02g031960.2J*, *BolC03g026110.2J*), cytokinin (*BolC03g061170.2J*), jasmonate (*BolC08g041070.2J*), and ABA (*BolC07g057090.2J*) pathways. While direct evidence connecting these specific regulators to *Fusarium* resistance remains elusive, their coordinated modulation suggests a complex hormonal crosstalk during defense activation. In the MAPK signaling cascade, differential gene regulation was observed: MEKK kinase (*BolC03g034380.2J*) was downregulated, while ACC synthase (*BolC08g044590.2J*) and defensins (*BolC06g030650.2J*) were significantly induced. Notably, *BolC06g030650.2J* encodes a PDF1.2 ortholog, a well-characterized biomarker of JA-mediated systemic acquired resistance (SAR) in crucifers, suggesting the evolutionary conservation of defense mechanisms across Brassicaceae family.

### 2.4. Stable Overexpression of PDF1.2 in Cabbage Enhances Resistance Against Foc

Owing to the conserved activation of JA signaling in crucifer–pathogen interactions [1,19], we focused on the functional characterization of *BolC06g030650.2J*, annotated as a *PDF1.2* ortholog (see Section 2.4). Using the *BolC06g030650.2J* coding sequence annotated in the cabbage reference genome, we designed gene-specific primers to amplify and sequence this locus in both YR01_20 and S01_20 near-isogenic lines. Comparative sequence analysis revealed complete conservation of the *PDF1.2* coding region between the two lines, with no identifiable single-nucleotide polymorphisms (SNPs), insertions, or deletions (Figure 4A). RT-qPCR analysis showed a 17.4-fold in the expression of *PDF1.2* in the resistant line at 12 hpi compared to the susceptible line (Figure 4B). Notably, the differential expression of *PDF1.2* observed between YR01_20 and S01_20 following *Foc* inoculation likely stems from divergent defense signaling cascades rather than differences in the intrinsic coding sequence (Figure 4B).

To examine its functional role, the 237 bp coding sequence (CDS) was directionally cloned into pCAMBIA1300-GFP, generating a C-terminal GFP fusion under the control of CaMV 35S promoter. *Agrobacterium tumefaciens* strain GV3101-mediated transformation of 800 hypocotyl explants from susceptible cultivar ‘YL’ yielded 6 putative transformants on MS medium supplemented with glufosinate. Genomic PCR confirmation using *Bar*-specific primers showed a transformation efficiency of 33% (2/6 lines), which were designated *OE-1* and *OE-2* (Figure 4C). Transcript accumulation analysis revealed over 3,000-fold overexpression in T0 plants (Figure 4D), and Western blot analysis confirmed the presence of the expected 35 kDa protein (Figure 4E). These results indicate the accuracy of the overexpression lines.

Pathogenicity assays conducted on T2 homozygous lines showed significantly enhanced resistance, with disease indices of 26.67 (*OE-1*) and 33.33 (*OE-2*), compared to 100.00 in wild-type controls at 12 days post-inoculation (dpi) (Figure 4F,G). Transgenic plants exhibited limited vein chlorosis and maintained structural integrity, whereas wild-type controls displayed complete systemic necrosis, suggesting that *PDF1.2* plays an important role in regulating *Fusarium* wilt resistance in cabbage.

## 3. Discussion

Our multi-omics integration reveals complex phytohormone interplay in cabbage’s defense against *Foc*, with jasmonates (JAs) and salicylic acid (SA) constituting core defense phytohormones, while cytokinins (CKs), abscisic acid (ABA), and ethylene (ET) exhibit modulatory roles in immune signaling networks.

### 3.1. Cytokinin Signaling in Foc Defense

The CK signaling cascade is initiated by membrane-localized histidine kinase receptors (HKs), transduced through histidine phosphotransfer proteins (HPs), ultimately activating response regulators (RRs) to orchestrate defense gene expression [25,26]. Our identification of three CK-related DEGs, two upregulated (*BolC03g044440.2J* [AHP], *BolC07g046020.2J* [B-ARR]) and one downregulated (*BolC03g061170.2J* [A-ARR]), suggests dynamic CK signal rewiring (Table 2). Notably, *Arabidopsis* B-ARR homolog *ARR2* enhances *Pseudomonas syringae* resistance via TGA3-mediated PR1 activation [27], proposing an evolutionarily conserved CK-SA crosstalk mechanism potentially operative in crucifer defenses.

### 3.2. ABA Signaling Dynamics

The core components of ABA signaling have been clearly defined: ABA receptors PYRABACTIN RESISTANCE (PYR/PYL)/ABA receptor regulatory components (RCAR) inhibit type 2C protein phosphatases (PP2C), thereby activating SNF1-related protein kinase 2 (SnRK2) protein kinases SnRK2.2, SnRK2.3, and OST1/SnRK2.6 [28]. However, ABA signaling components showed paradoxical regulation: three ABA receptor PYR/PYLs were suppressed (*BolC01g011300.2J*, *BolC03g024500.2J*, *BolC04g064360.2J*) while PP2C phosphatase (*BolC08g057620.2J*), a negative regulator, was induced (Table 2). This apparent contradiction mirrors the dual ABA roles reported in *Arabidopsis*—where *PYR1* overexpression enhances *Alternaria* resistance [29] yet *pyr1* mutants show improved *P*. *syringae* defense [30]. Recent study have shown that the PP2C member AIP1 (AKT1-interacting protein phosphatase 1) can interact with RCAR-type ABA receptors and may also be a positive regulator of ABA [31]. The coordinated PYR/PYL suppression and PP2C induction in resistant lines may establish an ABA signaling rheostat, potentially priming defense responses through SnRK2 kinase activation.

### 3.3. Growth-Defense Tradeoffs via SAUR/PIFs

The auxin-responsive SAUR (*BolC03g026110.2J*) and phytochrome-interacting factor PIF (*BolC02g031960.2J*) genes underscore the balance between growth and plant disease defense (Table 2). Most SAUR genes are involved in cell elongation during seedling development, while certain SAUR members induce senescence through interactions with PP2C.D phosphatases [32]. The knockout of *PIF4* significantly inhibits growth but enhances resistance to the bacterial pathogen *Pst* DC3000, whereas overexpression of *PIF4* promotes growth but compromises disease resistance [33]. This differential regulation suggests a reallocation of resources from growth to defense mechanisms in cabbage challenged by *F*. *oxysporum*.

### 3.4. BOP-Mediated Defense Convergence

*BolC08g041070.2J* encodes a BLADE ON PETIOLE (BOP) protein containing BTB/POZ and ankyrin repeat domains (Table 2). In *Arabidopsis*, there are six BTB-ankyrin proteins [34]. Among these, NPR1, NPR3, and NPR4 dominate plant defense responses through the SA signaling pathway [35,36,37], while NPR2 plays an auxiliary role in SA perception [38]. BOP1 and BOP2 are two other BTB-ankyrin proteins that play important roles in leaf and inflorescence structure development [39,40]. Additionally, *BOP1* and *BOP2* participate in defense regulation through the JA signaling pathway [41]. In cotton, *GhBOP1* enhances *Verticillium* resistance via lignin biosynthesis [42], suggesting a parallel mechanism where cabbage *BOP* may reinforce cell walls against *Foc* invasion.

### 3.5. CAPE1-like Immune Priming

*BolC07g057090.2J* encodes a CAPEs protein (Table 2). CAPEs are signaling peptides generated by the proteolytic cleavage of CAPs proteins and belong to the cysteine-rich secretory protein, antigen 5, and pathogenesis-related protein 1 (CAP superfamily). *PR1* is a member of the CAP superfamily and is widely present in various plants, exhibiting antimicrobial activity [43]. Mass spectrometry analysis identified an 11-residue peptide, CAPE1, derived from the C-terminus of PR1. *CAPE1* can induce reactive oxygen species (ROS) production in tomato leaves, activate stress responses and innate immunity, and enhance resistance to *P*. *syringae* without triggering cell death [44]. These properties of CAPE1 further suggest that CAPEs proteins may play a key role in plant defense mechanisms, particularly in regulating disease defense responses during pathogen infection.

### 3.6. MAPK-ET Crosstalk

In *Arabidopsis*, the recognition of the bacterial elicitor flg22 by the receptor kinase *FLS2* activates the MEKK1-MKK4/MKK5-MPK3/MPK6-WRKY22/WRKY29 cascade [45,46]. In this study, three genes involved in the MAPK signaling pathway were identified.. Among these, *BolC03g034380.2J* encodes a member of the MEKK (MAPK/ERK kinase kinase) family A1 subgroup. Unlike the cascade activated by *FLS2* recognition of flg22 in *Arabidopsis*, *BolC03g034380.2J* was significantly downregulated in resistant line post-inoculation (Table 2), suggesting the presence of another MAPK cascade pathway in cabbage resistance to *Foc*. The other two genes related to the MAPK signaling pathway are *BolC08g044590.2J* and *BolC06g030650.2J* (Table 2). *BolC08g044590.2J* encodes an ACC synthase, which converts S-adenosylmethionine (SAM) into 1-aminocyclopropane-1-carboxylate (ACC), and ACC is further converted into ethylene (ET) by ACC oxidase [47]. In *Arabidopsis*, *AtACS7*, *AtACS9*, and *AtACS11* maintain the balance between ET, ROS, and brassinosteroid hormones [48], while *AtACS2* and *AtACS5* participate in the ABA response pathway and control plant growth and development [49]. The up-regulated *BolC08g044590.2J* may drive ET accumulation to potentiate JA signaling while maintaining redox homeostasis.

### 3.7. The Role of PDF1.2 in Resistance to Foc

*BolC06g030650.2J* encodes a plant defensin family protein, and its homologous gene *PDF1.2* is a key gene in JA-mediated defense responses. The expression of *PDF1.2* depends on *COI1* and plays a positive regulatory role in plant resistance to pathogens [50]. Studies have shown that the JA and ET signaling pathways must work synergistically to activate *PDF1.2* expression during infection by the necrotrophic fungus *A*. *brassicicola* [51]. Additionally, in *Arabidopsis*, the *PDF1.2* promoter can be systemically activated by fungal pathogens and responds to methyl jasmonate (MeJA) [50,52]. Analysis of *PDF1.2* expression at multiple time points post-inoculation with *Foc* in near-isogenic lines showed that its expression pattern was consistent with the transcriptomic data (Figure 4B), validating the reliability of the data. Furthermore, this gene consistently exhibited significant differential expression between resistant and susceptible lines. This difference may be due to variations in the *PDF1.2* promoter sequence between resistant and susceptible lines or the regulation of *PDF1.2* expression by the resistance gene *FOC1* through unknown pathways. Although overexpression of *PDF1.2* significantly enhanced resistance to *Foc* (Figure 4F,G), the specific roles of the JA/ET signaling pathways in cabbage resistance to *Fusarium* wilt remain unclear. In summary, this finding provides an important clue for understanding the mechanisms of cabbage defense against *Foc* infection, particularly highlighting the potential key role of the JA/ET signaling pathway in defense responses, and points the way for future research.

## 4. Materials and Methods

### 4.1. Fungal Strain and Plant Material

*Foc* strain FGL03-6 (race 1), isolated from symptomatic cabbage plants in Yanqing, Beijing, China, was maintained through single-spore purification as described [5].

Development of near-isogenic lines with differential *Fusarium* wilt resistance: The early maturing cabbage cultivar ‘Viking Early Strain’ was introduced from Canada in 1966. Through six generations of selfing and single-plant selection, a genetically stabilized inbred line designated S01_20 was established. During 2009–2012, leveraging the *Fusarium* wilt-resistant cultivar ‘Zhonggan 18’, researchers generated 230 doubled haploid (DH) lines via the isolated microspore culture technique [6,53]. Beginning in 2012, the highly resistant DH line D134 was selected as the donor parent and subjected to three successive backcross generations with the elite recipient parent S01_20, followed by two selfing generations. This process successfully developed the resistant inbred line YR01_20, which exhibits agronomic traits nearly identical to those of S01_20 while maintaining robust *Fusarium* wilt resistance. Genetic similarity analysis confirmed that YR01_20 shared 99.8% genomic identity with the recurrent parent S01_20 [54].

Both the susceptible line S01_20 and the resistant line YR01_20 were provided by the Cabbage Breeding Group at the Institute of Vegetables and Flowers, Chinese Academy of Agricultural Sciences (IVF-CAAS). Plants were grown under controlled conditions (16 h light at 25 °C/8 h dark at 16 °C) until the three-leaf stage.

### 4.2. Pathogen Inoculation and Assessment

Root-dip inoculation was performed according to Lv et al. [5] with the following modifications. Fungal inoculum (1 × 10^6^ spores/mL) was applied to roots for 15 min. Disease severity was assessed 12 days post-inoculation using the following scale: 0 = no symptoms; 1 = slight yellowing of one leaf; 2 = moderate yellowing of one or two leaves; 3 = severe yellowing and/or wilting of half the leaves; 4 = severe yellowing or wilting of all leaves except the heart leaf; 5 = all leaves yellowed, severely wilted, or the plant death. The disease index (DI) was calculated as: [(Σ (disease grade × number of plants at corresponding grade)/(total number of plants × highest disease grade)] × 100.

### 4.3. RNA Extraction, cDNA Library Construction, and Sequencing

Root tissue samples from three independent biological replicates of each line (each replicate containing 3 plants) were collected at 0, 6, 12, 24, 48, 96, and 144 h post-inoculation, immediately frozen in liquid nitrogen, and stored at −80 °C for subsequent RNA-seq and qRT-PCR analysis. Total RNA was extracted from root tissues using the Plant RNA Kit (Tiangen, Beijing, China) according the manufacturer’s instructions, and genomic DNA was removed by DNase I treatment. RNA concentration and purity were assayed using an NanoDrop™ 1000 spectrophotometer (Thermo Fisher Scientific, Wilmington, DE, USA). Only RNA samples with an RNA integrity index (RIN) greater than 8.0, OD260/230 ratios between 2.0 and 2.5, and OD260/280 ratios between 1.9 and 2.1 were used for subsequent experiments. mRNA was enriched using oligo(dT) magnetic beads. First-strand cDNA was synthesized from the enriched mRNA using random hexamer primers, followed by second-strand cDNA synthesis. cDNA samples were purified, then end-repaired, tailed with a single adenine nucleotide, and ligated to sequencing adapters. cDNA fragments were size selected and enriched by PCR. All libraries were tested for quality control using an Agilent 2100 Bioanalyzer (Agilent Technologies, Santa Clara, CA, USA) and finally sequenced on the Illumina HiSeq^TM^ 2500 (Illumina, San Diego, CA, USA) platform in high-throughput mode. The RNA-seq data analyzed in this study are deposited in the NCBI SRA database (BioProject: PRJNA1231707).

### 4.4. Bioinformatics Analysis

Following RNA extraction, purification, and library preparation, paired-end sequencing was conducted on the Illumina HiSeq platform. Raw sequencing reads underwent quality control using Cutadapt (v3.4; http://code.google.com/p/cutadapt/, accessed on 4 September 2024) [55] to remove 3′-adapter sequences and filter low-quality reads (Phred score < Q20). The FastQC software (v0.11.9; http://www.bioinformatics.babraham.ac.uk/projects/fastqc/, accessed on 10 September 2024) [56] was utilized to analyze the single-base quality distribution of the sequencing data. High-quality reads were aligned to the *Brassica* reference genome (http://brassicadb.cn/, accessed on 14 September 2024) [21] via HISAT2 (v2.2.1; http://ccb.jhu.edu/software/hisat2/index.shtml, accessed on 19 September 2024) [22].

Gene expression quantification was performed using HTSeq (v0.13.5; http://www-huber.embl.de/HTSeq, accessed on 25 September 2024) [23] to generate read counts, which were normalized as FPKM (fragments per kilobase of transcript per million mapped reads) values [57] for cross-sample comparability. Genes with FPKM > 1 were classified as expressed, and their expression distributions were analyzed across intervals, with shared/unique genes visualized through UpSet plots. In the correlation analysis, Pearson correlation coefficients for all sample pairs were calculated based on FPKM-normalized expression values using the cor() function in R. The correlation coefficient matrix was visualized as a hierarchically clustered heatmap via the Pheatmap package (v1.0.12) [58].

To elucidate the molecular responses to *Foc* infection, comparative transcriptomics was performed at two critical infection time points (0 h and 12 hpi), selected based on preliminary pathogenicity assays, to delineate molecular dynamics at these stages. Four strategic comparisons were established: (i) genetic background variation between the susceptible and resistant lines under mock conditions (S01_20_MOCK vs. YR01_20_MOCK), (ii) transcriptional response of the susceptible line upon infection (S01_20_MOCK vs. S01_20), (iii) transcriptional response of resistant line upon infection (YR01_20_MOCK vs. YR01_20), and (iv) genotype-specific resistance response at 12 hpi (S01_20 vs. YR01_20). Differential expression analysis was conducted using DESeq2 (v1.30.1) [24] with significance thresholds set at FDR < 0.05 and |log2FC| > 1. Volcano plots (generated with ggplot2 v3.3.5 [59]) illustrated fold changes and statistical significance of DEGs, while genomic annotations of DEGs were mapped using Circlize (v0.4.13). Shared and unique DEGs across comparisons were quantified via UpSet plots, and a union set of all DEGs underwent bidirectional hierarchical clustering (pheatmap) based on Euclidean distances and complete linkage.

Functional enrichment analysis included GO term identification using topGO (hypergeometric test, *p* < 0.05) [60] to highlight DEG-associated biological processes, and KEGG pathway evaluation using Rich factor (ratio of DEGs to total annotated pathway genes) and FDR values, with lower FDR values indicating stronger enrichment [61].

### 4.5. Real-Time Quantitative PCR Analysis

Total RNA was extracted using an RNA extraction kit, as detailed in Section 4.2. Subsequently, first-strand cDNA was synthesized using the Transcriptor First-Strand cDNA Synthesis Kit (Roche, Indianapolis, IN, USA), adhering strictly to the manufacturer’s guidelines. The experimental protocol was adapted from Liu et al. [1] with minor modifications. Real-time quantitative PCR (RT-qPCR) was performed using the CFX96 Touch™ Real-Time PCR System (Bio-Rad Laboratories, Hercules, CA, USA). A 20 μL RT-qPCR reaction mixture was prepared, comprising 10 μL of SYBR Green PCR Master Mix (Vazyme Biotech Co., Ltd., Nanjing, China), 1 μL of cDNA, 0.5 μL each of the forward and reverse primers, and ddH_2_O to bring the final volume to 20 μL. Each sample was assayed in triplicate. The relative expression levels of the DEGs were calculated using the 2^−∆∆Ct^ method [62,63], with actin serving as the internal reference gene. *Actin* (GenBank accession number XM_013731369.1; RT-qPCR-F: 5′-CCAGAGGTCTTGTTCCAGCCATC-3′; RT-qPCR-R: 5′-GTTCCACCACTGAGCACAATGTTAC-3′) was used as the internal control gene to normalize the expression of the target genes. The quantitative primers (RT-qPCR-F: 5′-TCTCTTTGCTGCTCTTGTTTTC-3′; RT-qPCR-R: 5′-TGTGAGCAGGGAAGACATAATT-3′) were designed based on the coding sequence of *BolC06g030650.2J* using Primer 6.0. Data processing and graphical representation were executed using GraphPad Prism 10.0 (GraphPad Software, San Diego, CA, USA).

### 4.6. Plasmid Construction and Agrobacterium-Mediated Plant Genetic Transformation

To validate the disease resistance function of *BolC06g030650.2J*, a 237 bp fragment (excluding the stop codon) was amplified from the cDNA library of YR01_20 and cloned into a modified pCAMBIA1300 vector. The sequences of the amplification primers (containing *BamH*I restriction sites in OE-F and *Sac*I in OE-R) are provided below: BolC06g030650.2J-OE-F: 5′-CTCTCTCTCAAGCTTGGATCCATGGCTAAGGCTGCTACCATCA-3′; BolC06g030650.2J-OE-R: 5′-ACGGGTCATGAGCTCCTGCAGACATGGGAAGTAGCAGATGCAC-3′. This plasmid contained *Bar* and *Kana* resistance genes for selection purposes during *Agrobacterium*-mediated transformation. All plasmids underwent Sanger sequencing verification at Shanghai Sangon Biotech Co., Ltd. (Shanghai, China) and were subsequently transformed into *A. tumefaciens* strain GV3101.

For cabbage genetic transformation, hypocotyls from 6-day-old seedlings were excised into 0.8–1 cm explants, pre-cultured for 2 days, and then inoculated and co-cultured with *A. tumefaciens* strain GV3101 carrying the target vector. Following 2 days of co-cultivation, the explants were transferred to selection medium (composition: 4.43 g/L MS, 0.1 mg/L 1-naphthaleneacetic acid, 1 mg/L 6-benzylaminopurine, 30 g/L sucrose, 12 mg/L Basta, 300 mg/L Timentin (inhibition of *Agrobacterium* overgrowth), and 0.8% agar; pH adjusted to 5.84–5.90). Under these conditions, Basta-resistant shoots were selected. Transgenic plants were confirmed by PCR amplification using primers specific to *Bar*. The primer sequences employed for the detection of overexpression materials are as follows: Bar-F: 5′-CAGCTGCCAGAAACCCACGTCATG-3′; Bar-R: 5′-GGTCAACTTCCGTACCGAGCCGC-3′. The experimental protocol described above was refined through minor adjustments and further developed based on the methodology established by Han et al. [64].

### 4.7. Protein Extraction and Western Blotting

Freshly harvested *OE-1/2* leaves were placed into a mortar that has been pre-cooled with liquid nitrogen. Total proteins were subsequently extracted using a buffer consisting of 100 mM Tris (pH 7.5), 150 mM NaCl, 1 mM EDTA (pH 8.0), 0.5% Nonidet P-40, and 1 mM dithiothreitol. The crude protein extract was placed horizontally on ice in a shaker and agitated at 100 rpm/min for 30 min. After agitation, it was centrifuged at 12,000 rpm/min for 20 min using a pre-cooled centrifuge. The resulting supernatant was then transferred to a new centrifuge tube. An aliquot of the supernatant was pipetted and mixed with sample loading buffer, heated to 95 °C for 10 min to facilitate elution, and then separated on a 15% Tris-Gly SDS-PAGE gel. The separated proteins were subsequently transferred onto a PVDF (polyvinylidene fluoride) membrane. Following transfer, nonspecific binding sites on the membrane were blocked with skimmed milk. The membrane was then incubated overnight with the primary antibody (mouse anti-GFP, Roche). After incubation with the primary antibody, unbound primary antibody was removed using 1× TBST buffer. Subsequently, a secondary antibody conjugated with HRP (goat anti-mouse horseradish peroxidase, Sigma-Aldrich) was added for incubation. After incubation with the secondary antibody, unbound secondary antibody was removed using 1× TBST buffer. Finally, ECL chemical luminescence solution (ECL Substrate Kit, Abcam) was added, and the target protein bands were detected using a chemiluminescence detector. The experimental protocol described above was refined with minor modifications based on the research conducted by Guo et al. [65].

## 5. Conclusions

In this study, a comparative transcriptome analysis was performed using RNA-seq on NILs with contrasting resistance phenotypes at multiple time points post-inoculation. The results showed that the plant hormone signal transduction pathway and the plant MAPK signaling pathway were the most significantly enriched, containing the highest number of DEGs. Notably, the gene *BolC06g030650.2J*, encoding a plant defensin family protein homologous to PDF1.2 (a well-characterized marker for JA-mediated defense responses), was significantly upregulated in the resistant line challenged with *Foc*. Through genetic transformation and artificial inoculation experiments in cabbage, we verified the positive regulatory role of *PDF1.2* in enhancing cabbage resistance to *Fusarium* wilt.

## Figures and Tables

**Figure 1 ijms-26-03770-f001:**
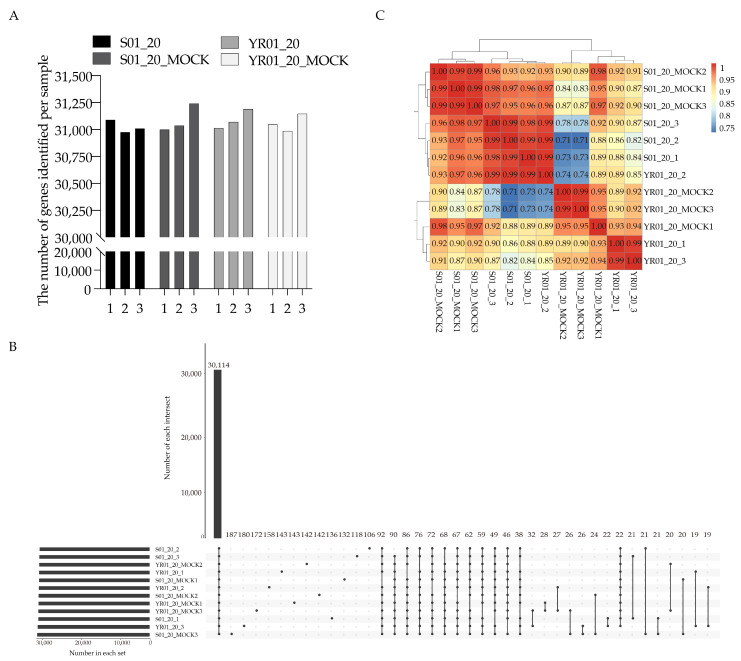
Gene expression analysis and sample correlation. (**A**) Total number of expressed genes identified for each sample. Biological triplicates (labeled 1–3) are shown for each group, with gene expression defined by the FPKM > 1 cutoff. (**B**) UpSet plot analysis of shared and unique expressed genes across samples. Bar heights represent the total number of expressed genes per sample, while intersection nodes (connected lines) indicate genes shared among multiple samples. The baseline spanning all samples on the x-axis represents universally detected genes. (**C**) Heatmap of pairwise Pearson correlation coefficients among samples. Hierarchical clustering (top/left margins) groups samples based on expression similarity, with correlation coefficients (color scale: 0.8–1.0 indicating strong correlation). Coefficients below 0.8 between replicates indicate suboptimal reproducibility.

**Figure 2 ijms-26-03770-f002:**
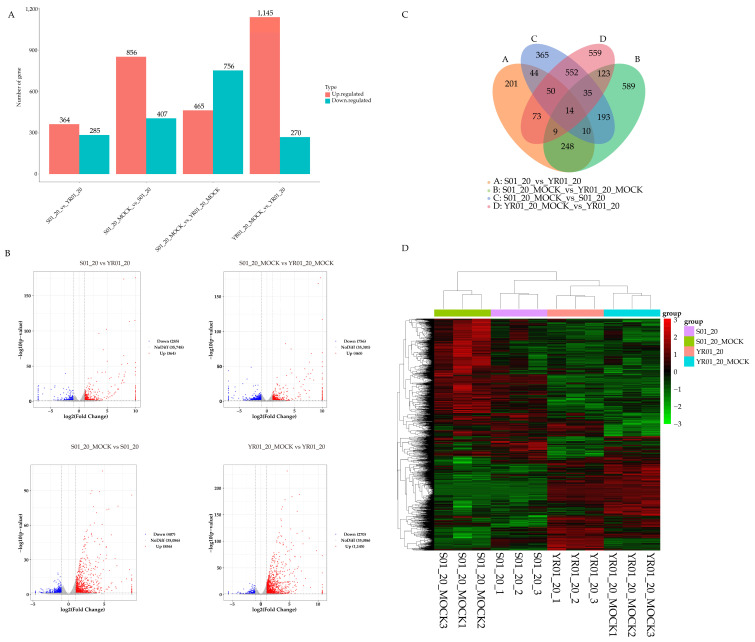
Differential expressed genes (DEGs) profiling across experimental groups. (**A**) Bar plot showing the number of significantly upregulated (red) and downregulated (blue) genes in pairwise comparisons between resistant and susceptible lines. DEGs were identified using a threshold of |log2FoldChange| > 1 and *p* < 0.05. (**B**) Volcano plot showing the distribution of DEGs. The x-axis represents log2-transformed fold changes, and the y-axis displays −log10-transformed *p*-values. Dotted lines indicate significance thresholds (|log2FC| = 1, *p* = 0.05). Data points are color-coded: red (upregulated), blue (downregulated), and gray (non-significant). The symmetrical distribution of DEGs reflects expected expression divergence across conditions. (**C**) Venn diagram showing shared and unique DEGs among comparison groups. Overlapping regions indicate conserved DEGs among comparison groups, while non-overlapping sections represent context-specific gene sets. (**D**) Heatmap showing hierarchical clustering of DEGs expression patterns. The rows (genes) and columns (samples) are hierarchically clustered using the Euclidean distance with complete linkage. Expression levels are normalized (red: high; green: low), revealing co-regulated gene modules and sample subgroup consistency.

**Figure 3 ijms-26-03770-f003:**
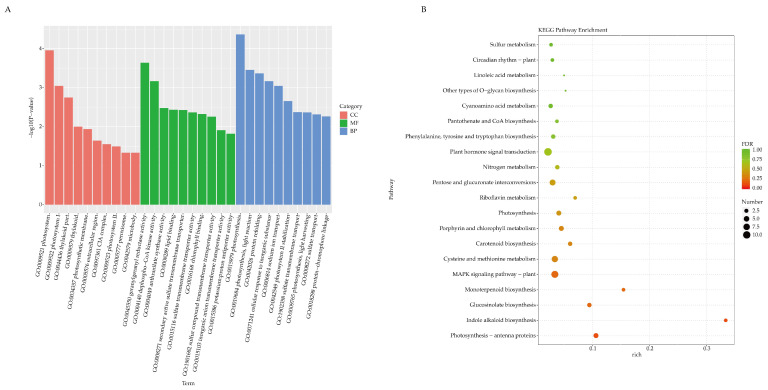
Functional enrichment analysis of DEGs in S01_20 vs. YR01_20. (**A**) Bar plot showing the top 10 significantly enriched terms in the three categories of cellular component, biological processes, and molecular function for DEGs in S01_20 versus YR01_20. Bar height represents the enrichment factor [(Number of DEGs in term)/(Number of background genes in term)]. Terms with FDR-adjusted *p*-value < 0.05 (Benjamini–Hochberg method) were considered significant, with level 2 annotations highlighting broad functional classifications. (**B**) The bubble plot visualizes the top 20 significantly enriched KEGG pathways. The x-axis represents the enrichment factor (calculated as [DEGs in pathway]/[background genes in pathway]), while the y-axis displays pathway nomenclature sorted by descending enrichment factor. Bubble diameter correlates with the number of DEGs annotated per pathway. A color gradient from red to yellow reflects ascending FDR thresholds (crimson: FDR < 0.01; amber: FDR < 0.05; pale yellow: FDR ≥ 0.05), with color intensity scaled proportionally to −log10(FDR) values. Gray dashed lines demarcate the conventional significance threshold (FDR = 0.05, Benjamini–Hochberg adjusted).

**Figure 4 ijms-26-03770-f004:**
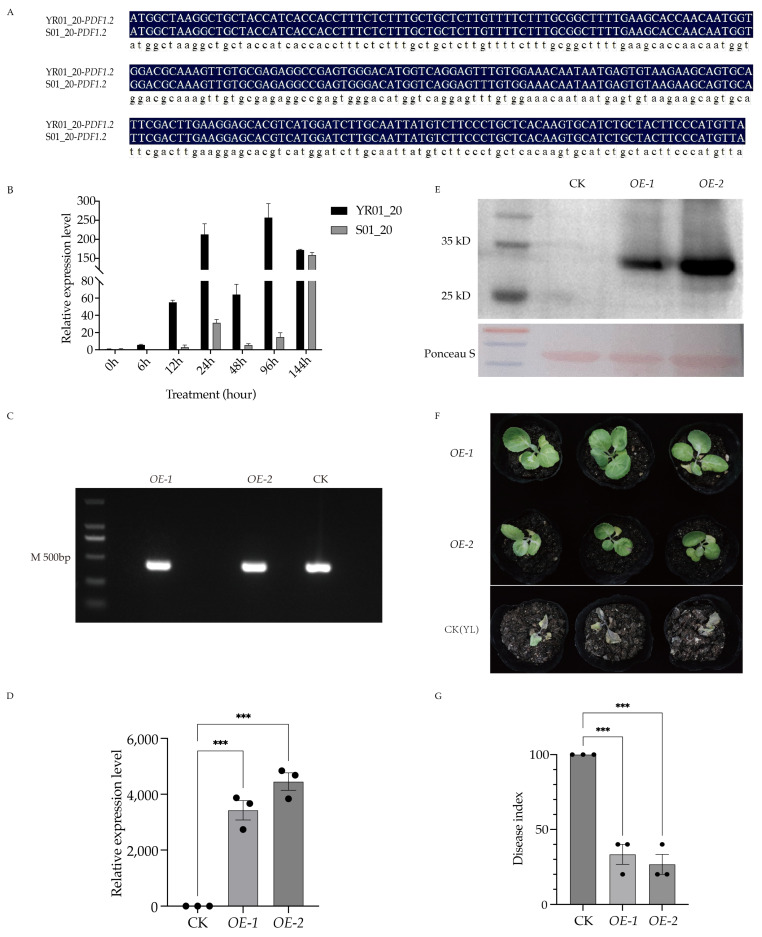
Functional characterization of *PDF1.2* in disease response. (**A**) Sequence alignment analysis of *PDF1.2* in NILs. (**B**) Time-course expression profiling of *PDF1.2* in NILs following *Foc* inoculation. (**C**) Genotypic validation of *PDF1.2* overexpression (*OE*) lines via transgene-specific PCR. (**D**) RT-qPCR confirmation of *PDF1.2* transcript levels in *OE* lines. (**E**) Western blot analysis of PDF1.2 protein accumulation in *OE* lines. (**F**) Pathogen resistance assay comparing disease progression in *OE* and control lines after artificial inoculation. Asterisks denote statistical significance levels: *** corresponds to *p* < 0.001. (**G**) Disease severity quantification in *OE* lines, expressed as a symptom index (0–100% scale).

**Table 1 ijms-26-03770-t001:** Statistics of the sequencing data.

Sample ID	Reads No.	Bases (bp)	Clean Reads No.	Clean Data (bp)	Clean Reads (%)	Q20 (%)	Q30 (%)
S-Foc-1	44,603,366	6,735,108,266	41,427,642	6,255,573,942	92.88	97.54	94.22
S-Foc-2	43,900,734	6,629,010,834	40,789,702	6,159,245,002	92.91	97.52	93.95
S-Foc-3	42,642,538	6,439,023,238	39,572,722	5,975,481,022	92.8	97.69	93.79
S-MOCK1	39,137,790	5,909,806,290	36,291,182	5,479,968,482	92.72	97.65	94.06
S-MOCK2	39,511,534	5,966,241,634	36,646,568	5,533,631,768	92.74	97.48	93.93
S-MOCK3	48,918,954	7,386,762,054	45,412,050	6,857,219,550	92.83	97.41	94.3
YR-Foc-1	39,549,064	5,971,908,664	36,671,228	5,537,355,428	92.72	97.54	94.08
YR-Foc-2	41,756,658	6,305,255,358	38,804,600	5,859,494,600	92.93	97.01	93.94
YR-Foc-3	49,111,204	7,415,791,804	45,588,484	6,883,861,084	92.82	97.1	93.97
YR-MOCK1	39,428,946	5,953,770,846	36,585,610	5,524,427,110	92.78	97.58	93.98
YR-MOCK2	39,042,744	5,895,454,344	35,950,636	5,428,546,036	92.08	97.51	93.37
YR-MOCK3	41,046,088	6,197,959,288	37,813,296	5,709,807,696	92.12	97.55	93.56
Total	508,649,620	76,806,092,620	471,553,720	71,204,611,720	92.71	97.47	93.94

Note: Sample IDs are as follows: S-Foc represents the treatment group from susceptible line S01_20, while S-MOCK denotes its matched control group (S01_20_MOCK); similarly, YR-Foc corresponds to the treatment group from the NIL resistant line YR01_20, and YR-MOCK designates its control counterpart (YR01_20_MOCK). The terms used in the analysis are defined as follows: Reads No.: the total number of paired-end reads; Bases (bp): the total number of bases; Clean Reads No.: the total paired-end reads retained after quality filtering processes; Clean Data (bp): the cumulative nucleotide count within the filtered dataset; Clean Reads (%): the proportion of high-quality reads (see text for definition) relative to the original raw sequencing output; Q20 (%): the percentage of bases with a Phred quality score ≥ 20, corresponding to ≥99% base-calling accuracy; Q30 (%): the percentage of bases with a Phred score ≥ 30 (≥99.9% accuracy).

**Table 2 ijms-26-03770-t002:** The list of DEGs enriched in plant hormone signal transduction and the MAPK signaling pathway.

Pathway	Gene	Expression	Annotation
Plant hormone signal transduction	*BolC03g044440.2J*	Up	Encodes histidine phosphate transferase, cytokinin
	*BolC07g046020.2J*	Up	Encodes an Arabidopsis response regulator (ARR) protein, cytokinin
	*BolC08g057620.2J*	Up	Encodes a member of the PP2C family, abscisic acid
	*BolC07g057090.2J*	Down	Encodes a CAP (cysteine-rich secretory proteins, antigen 5, and pathogenesis-related 1 protein) superfamily protein, salicylic acid
	*BolC01g011300.2J*	Down	Encodes a PYR/PYL/RCAR family protein, abscisic acid
	*BolC02g031960.2J*	Down	SAUR-like auxin-responsive protein family, auxin
	*BolC03g061170.2J*	Down	Encodes a transcription repressor that mediates a negative feedback loop in cytokinin signalling, cytokinin
	*BolC03g026110.2J*	Down	Encodes a nuclear localized bHLH protein, auxin
	*BolC03g024500.2J*	Down	Encodes a PYR/PYL/RCAR family protein, abscisic acid
	*BolC04g064360.2J*	Down	Encodes a PYR/PYL/RCAR family protein, abscisic acid
	*BolC08g041070.2J*	Down	Encodes a protein containing ankyrin repeat domains and BTB/POZ domains, jasmonic acid
MAPK signaling	*BolC06g030650.2J*	Up	Encodes an ethylene- and jasmonate-responsive plant defensin, belongs to the plant defensin (PDF) family
	*BolC08g044590.2J*	Up	Encodes a member of the 1-aminocyclopropane-1-carboxylate (ACC) synthase gene family
	*BolC03g034380.2J*	Down	Encodes a member of the A1 subgroup of the MEKK (MAPK/ERK kinase kinase) family

Note: The functional annotations of DEGs listed in Table 2 were derived from The *Arabidopsis* Information Resource (TAIR; https://www.arabidopsis.org), the authoritative genome database for *Arabidopsis thaliana*.

## Data Availability

All data supporting the findings of this study are available in this paper and its Appendix A, published online.

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
