# Peer review of "Transcriptome Analysis of Cabbage Near-Isogenic Lines Reveals the Involvement of the Plant Defensin Gene PDF1.2 in Fusarium Wilt Resistance"

_ijms, 2025, doi:10.3390/ijms26083770_

Round 1

Reviewer 1 Report

Comments and Suggestions for Authors

The manuscript reported a comprehensive RNAseq analysis on cabbage near-isogenic lines for Fusarium resistance. The study will be helpful for understanding the molecular mechanisms governing cabbage resistance to Fusarium. The results of target genes associated to the resistance are also be interested by the cabbage breeders. The feasibility and reliability of the molecular methods were clearly presented, and the figures are in good quality. It is suitable for the publication after minor revision.

  1. Materials of near-isogenic lines may have a detail description for how they were developed.
  2. Root-dip inoculation and the disease severity scored by leaves, however the RNAseq was only conducted from root. The reseason may be provided.
  3. All the scientific name of plant or fungus species, after it appeared first time, the genera name should be abbreviated.
  4. Whether the PDF1.2 gene contained some sequences variation between the two lines with resistance or susceptible. Is it used as marker for resistance survey in breeding practices.
  5. All the species names may be italic in Reference part.

Author Response

Thank you for your letter and for the reviewers' comments concerning our manuscript entitled “Transcriptome analysis of cabbage near-isogenic lines reveals the involvement of the plant defensin gene PDF1.2 in Fusarium wilt resistance” (Manuscript ID: ijms-3541536). The reviewers' feedback was extremely valuable and helpful in revising and improving our paper, and it has provided important guidance for our research. We have carefully reviewed the comments and made the necessary corrections, which we hope will meet with your approval. Following your thoughtful suggestions, we have made extensive revisions to our previous draft, and the detailed corrections are listed below.

We sincerely thank the editor and all reviewers for their valuable feedback, which we have used to improve the quality of our manuscript. The reviewers' comments are presented below in italicized font, and specific concerns have been numbered. Our responses are given in regular font, and additions to the manuscript are highlighted in green text. Based on the editor’s and reviewers' comments, we have made extensive modifications to our manuscript and supplemented additional data to strengthen our results.

Thank you again for your constructive feedback and valuable suggestions, which have greatly improved the quality of our manuscript.

Reviewer 2 Report

Comments and Suggestions for Authors

The paper is generally well-presented and written but includes a few errors and unclear sentences. Specific examples are listed below. After these issues are addressed the reviewer assesses that the paper will be acceptable for publication in Int. J. Molec. Sci.

Abstract

Line 33) “…these finding not only advances …” change to “…these findings advance…”.

Introduction

Line 49) “…China belong…” suggest change to “…China thusfar belong…”. Failure to find a race doesn’t necessarily mean it isn’t present.

Line 50) “…marker development.” Suggest change to “…marker development to support breeding for resistance.”

Lines 53-54) “Trough map-based cloning, FOC1 resistance gene …” change to “Through map-based cloning the FOC1 resistance gene…”.

Lines 56-57) “…Chinese cabbage materials, FocBo1/FocBr1 was confirmed to be the same gene of FOC1.” Suggest change to “…Chinese cabbage germplasm, FocBo1/FocBr1 was confirmed to be isogenic with FOC1 [7,8].” Is this germplasm B. rapa or B. napus? If so, a note should be added to the text to denote.

Line 58) “…innate immune system comprises two lines of defense: upon…” change to “…disease defenses are comprised of two systems: (1) upon…”.

Line 59) “…the cell surface…” change to “…the plant cell surface…”.

Line 62) “…defense, effector-triggered immunity…” change to “…defense, (2) effector-triggered immunity…”.

Line 67) “…plant immunity…” suggest change to “…plant resistance…”. Many plant pathologists contend that plant disease resistance does not emulate immunity mechanisms present in animals and that the term “immunity” should not be used.

Line 69) “…over 80% of characterized resistance proteins.” Literature references are needed here for substantiation.

Line 71) “The Fusarium wilt resistance genes…” suggest new paragraph.

Line 71) “…successfully cloned in melon…” suggest change to “…have been successfully cloned from melon…”.

Line 72) “…(Brassica oleracea)…” delete.

Lines 73-74) “Activation of R genes triggers downstream immune signaling, involving mitogen-activated protein kinase (MAPK) signaling pathway…” Change to “…The activation of R genes triggers downstream defense signaling, involving the mitogen-activated protein kinase (MAPK) signaling pathway…”.

Line 87) “…with 9 genes showing significant upregulation and 5 exhibiting…” change to “…with nine genes showing significant upregulation and five exhibiting…”.

Line 88) “…function coordinatly…” change to “…function in coordination with…”.

Results

Line 102) “…respectively.” Delete.

Table 1) This table should be reformatted so that lines and columns numerical attributes are consistent.

Line 119) “…high-quality reads relative…” change to “…high-quality reads (see text for definition) relative…”.

Line 122) “…Brassica oleracea…” change to “…B. oleracea…”.

Line 142) “…resultsand…” change to “…results and…”.

Line 155) The paragraph starting on this line should be moved to Materials & Methods.

Line 170) “Volcano plot…” change to “A volcano plot…”.

Lines 173- 174 and 175) “…of DEGs…” delete (redundant).

Line 181) “…comparisons…” change to “…comparisons between resistant and susceptible nils…”.

Line 183) “…linesindicate…” change to “…lines indicate…”.

Line 194) “…are employed…” change to “…were employed…”.

Line 195) “Cellular component…” suggest change to “(i) Cellular component…”.

Line 196) “…, molecular function…” suggest change to “; (ii) molecular function…”.

Line 197) “…, and biological processes…” suggest change to “…;(iii) and biological processes…”.

Line 218) “…plant immunity…” suggest change to “…plant disease defense…”.

Line 234) “…across Brassicaceae family…” suggest change to “…across Brassicaceae…”.

Table 2) Minor formatting is needed to align terms in rows and columns.

Discussion

Line 303) “…immunity...” suggest change to “…plant disease defense…”.

Line 331) “…immune…” suggest change to “…disease defense…”.

Line 355) “…SA…” reviewer could not identify this acronym; do authors intend “…JA…”?

Materials & Methods

Authors should ensure that all coopted molecular methods are either described in detail or that citations to the methods are included (see, for example, “Cutadapt” below).

Authors should also change all present nouns pertaining to tasks related to performed procedures to the past tense. This section has many examples, some but not all described enumerated below.

Line 378) “…with modifications…” change to “…with the following modifications…”.

Line 408) “…using Cutadapt…” this procedure should be described or a citation to the method added to text.

Line 436) “…Real-Time quantitative…” change to new sentence.

Line 441) “…differentially expressed genes (DEGs)…” change to “…DEGs…”.

Line 481) “…PVDF…” define this acronym.

Line 481) “…membrane are…” change to “…membrane were…”.

Line 484) “…antibody is washed away using…” change to “…antibody was removed using…”.

Conclusions

Lines 491-493) “CFW is a highly destructive disease that leads to substantial yield losses in cabbage crops. Although the resistance gene FOC1 has been successfully mapped, the molecular mechanisms underlying FOC1-mediated resistance remain poorly understood.” These introductory sentences are redundant with the abstract and can be removed to serve the purposes of the Conclusion section.

Line 499) “…upregulated.” Change to “…upregulated in the resistant line challenged with FOC.”

Author Response

Thank you for your letter and for the reviewers' comments concerning our manuscript entitled “Transcriptome analysis of cabbage near-isogenic lines reveals the involvement of the plant defensin gene PDF1.2 in Fusarium wilt resistance” (Manuscript ID: ijms-3541536). The reviewers' feedback was extremely valuable and helpful in revising and improving our paper, and it has provided important guidance for our research. We have carefully reviewed the comments and made the necessary corrections, which we hope will meet with your approval. Following your thoughtful suggestions, we have made extensive revisions to our previous draft, and the detailed corrections are listed below.

We sincerely thank the editor and all reviewers for their valuable feedback, which we have used to improve the quality of our manuscript. The reviewers' comments are presented below in italicized font, and specific concerns have been numbered. Our responses are given in regular font, and additions to the manuscript are highlighted in yellow text. Based on the editor’s and reviewers' comments, we have made extensive modifications to our manuscript and supplemented additional data to strengthen our results.

Thank you again for your constructive feedback and valuable suggestions, which have greatly improved the quality of our manuscript.
